# Identification of a critical sulfation in chondroitin that inhibits axonal regeneration

Craig S Pearson[1,2], Caitlin P Mencio[1], Amanda C Barber[2], Keith R Martin[2]*, Herbert M Geller[1]*

[1]Laboratory of Developmental Neurobiology, National Heart, Lung, and Blood Institute, National Institutes of Health, Bethesda, United States; [2]Department of Clinical Neurosciences, University of Cambridge, Cambridge, United Kingdom

**Abstract** The failure of mammalian CNS neurons to regenerate their axons derives from a combination of intrinsic deficits and extrinsic factors. Following injury, chondroitin sulfate proteoglycans (CSPGs) within the glial scar inhibit axonal regeneration, an action mediated by the sulfated glycosaminoglycan (GAG) chains of CSPGs, especially those with 4-sulfated (4S) sugars. Arylsulfatase B (ARSB) selectively cleaves 4S groups from the non-reducing ends of GAG chains without disrupting other, growth-permissive motifs. We demonstrate that ARSB is effective in reducing the inhibitory actions of CSPGs both in in vitro models of the glial scar and after optic nerve crush (ONC) in adult mice. ARSB is clinically approved for replacement therapy in patients with mucopolysaccharidosis VI and therefore represents an attractive candidate for translation to the human CNS.

DOI: https://doi.org/10.7554/eLife.37139.001

*For correspondence:
krgm2@cam.ac.uk (KRM);
gellerh@nhlbi.nih.gov (HMG)

Competing interests: The authors declare that no competing interests exist.

## Introduction

There is an urgent need for therapies to treat CNS injuries. Acute insult often results in axonal degeneration, and therefore many experimental strategies aim to stimulate regeneration of damaged axons. These efforts have been informed by two insights: (1) the extrinsic environment of the adult CNS is hostile to axonal growth due to the formation of a glial scar, and (2) adult CNS neurons have lost their intrinsic ability to express axon-growth-promoting factors. The inhibitory properties of the glial scar are primarily due to extracellular matrix molecules, particularly chondroitin sulfate proteoglycans (CSPGs) (*Burnside and Bradbury, 2014*). Digesting CSPG GAG chains with the bacterial enzyme chondroitinase ABC (ChABC) has been shown to promote axonal extension in several experimental models (*Bradbury and Carter, 2011*; *Zhao and Fawcett, 2013*). In parallel, many approaches have endeavored to restore the intrinsic growth capability of CNS axons, most prominently through the activation of cell growth programs such as the PTEN/mTOR pathway (*Park et al., 2008*). However, such approaches are not directly translatable to human patients: ChABC has failed to reach clinical trials, and manipulation of tumor suppressor genes is likely to prove clinically questionable (*Barber et al., 2017*).

Increasing evidence supports a critical role for GAG chain sulfation in CSPG signaling. Global removal of sulfate groups from GAG chains eliminates the inhibitory actions of CSPGs in culture (*Smith-Thomas et al., 1995*), and specific sulfation motifs dictate whether GAGs inhibit or permit axon growth. For instance, axons grow readily over surfaces coated with 6-sulfated (6S) CSPGs (*Wang et al., 2008*), and deleting the enzyme that adds 6S to CS GAGs impairs axonal regeneration in mice (*Lin et al., 2011*). In contrast, axons avoid 4-sulfated (4S) CSPGs, an effect abolished by treatment with 4-sulfatase (*Wang et al., 2008*). Sulfation at both positions (4,6S) has been shown to

inhibit axonal growth in vitro and in vivo (*Brown et al., 2012*), and an increase in the ratio of 4S to 6S has been linked with age-related declines in plasticity (*Foscarin et al., 2017*; *Miyata et al., 2012*). Collectively, these observations suggest that reducing 4S while preserving 6S on intact GAG chains may enable growing axons to overcome CSPG-mediated inhibition more effectively than indiscriminate reductions in sulfation or destruction of GAGs.

We have previously shown that the deposition of CSPGs in the glial scar following brain and spinal cord injury is dominated by 4S GAGs (*Wang et al., 2008*). Here, we demonstrate the ability of arylsulfatase B (ARSB), a clinically approved enzyme that selectively removes 4S groups from the non-reducing ends of CSPGs (*Litjens and Hopwood, 2001*), to enhance axonal regeneration in vitro and in vivo. Treating CSPGs with ARSB or adding ARSB to TGF-β-treated astrocytes reverses their inhibition of neurite outgrowth by modifying CSPG sulfation. We then show that ARSB significantly enhances axon regeneration in vivo when the enzyme is delivered to the injured optic nerves of adult mice in combination with intravitreal injection of Zymosan and CPT-cAMP, an intrinsic growth stimulus. Importantly, this treatment is effective when administered several days after ONC, making it relevant for human conditions where interventions are rarely available immediately following injury. Crucially, ARSB (Naglazyme, Biomarin) is clinically approved for the treatment of mucopolysaccharidosis VI, a lysosomal storage disorder (*Muñoz-Rojas et al., 2010*; *Harmatz et al., 2004*; *Harmatz et al., 2005*); thus, its inclusion in future human therapeutic treatments is plausible. Taken together, these data establish a critical role for 4S at the non-reducing end of CS GAG chains in mediating the inhibitory actions of CSPGs. Moreover, they provide evidence for a promising translatable therapy that utilizes a highly selective human enzyme to modify chondroitin sulfation and enhance axon regeneration.

## Results

### ARSB reverses the inhibition of neurite growth caused by 4-sulfated CSPGs

The ability of ARSB to alter the inhibitory actions of CSPGs was first assessed in cell culture models of the glial scar (*Wang et al., 2008*). To assess whether neurite inhibition by CSPGs could be reduced through ARSB treatment, cultures of dissociated mouse hippocampal neurons were exposed to 5 µg/ml CSPGs with and without ARSB treatment for 48 hr. Cultures were stained for βIII-tubulin, and the lengths of neurites were measured. Neurons grown in the presence of CSPGs were significantly ($p<0.0001$) shorter than untreated neurons (neurite length [median]: 55.7 µm and 91.7 µm, respectively) (*Figure 1c–d*). Growth was not significantly altered by CSPGs that had been treated with ARSB (neurite length [median]: 93.2 µm), suggesting that ARSB treatment was sufficient to remove neurite outgrowth inhibiting characteristics of CSPGs (*Figure 1c–d*).

To test the actions of ARSB in a cellular model, monolayers of confluent mouse astrocytes were treated with TGF-β to stimulate elevated CSPG production (*Wang et al., 2008*). Mouse cerebellar granule neurons (CGNs) were then seeded onto these astrocytes and allowed to grow for 24 hr. Cultures were stained for GFAP and βIII-tubulin, and the lengths of CGN neurites were measured. Neurons growing on TGF-β-treated astrocytes exhibited significantly lower neurite outgrowth than those plated on untreated control astrocytes ($p=0.0059$, Mann-Whitney U test) (*Figure 1e–f*). However, incubating TGF-β-treated co-cultures with ARSB restored average neurite length to the levels observed in untreated controls (*Figure 1e–f*), significantly different from TGF-β-treatment alone ($p<0.0001$). This suggests that cleaving 4S from the non-reducing ends of GAG chains is sufficient to neutralize the inhibitory effects of CSPGs on neurons.

To demonstrate that ARSB acts on extracellular CSPGs, rather than being internalized into astrocytes and interfering with CSPG production or secretion, conditioned medium (CM) was collected from TGF-β-treated astrocytes and left untreated, treated with ARSB, or treated with ChABC. The isolated and treated CM was added to separately cultured CGNs. Application of CM from TGF-β-treated astrocytes significantly reduced neurite outgrowth while ARSB treatment reversed this effect to a degree equivalent to ChABC (*Figure 1—figure supplement 1*). Together, these findings demonstrate that the presence of CSPGs can inhibit neurite outgrowth, and that this inhibition is overcome by exposing the CSPGs to either ARSB or ChABC.

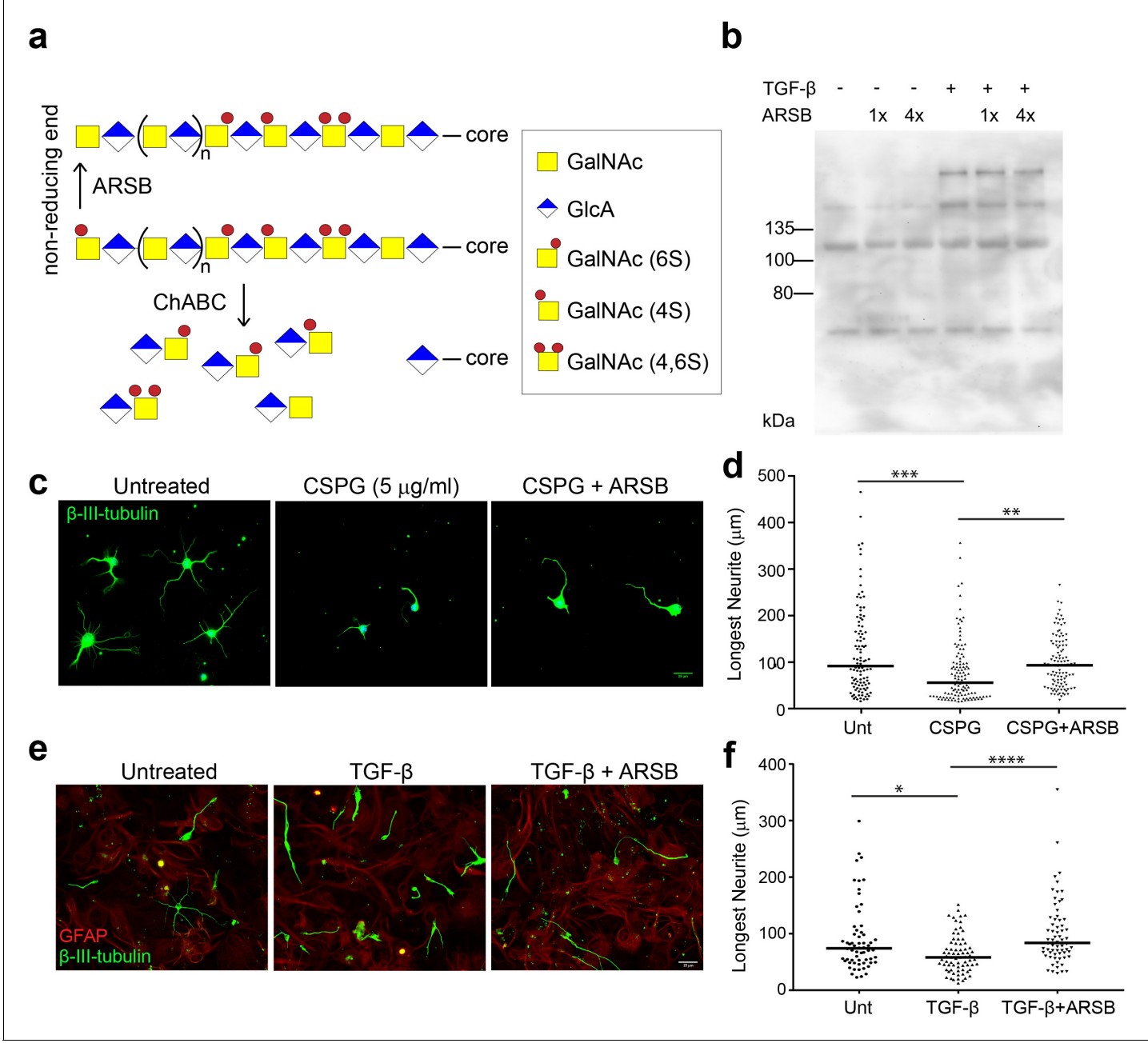

**Figure 1.** ARSB reverses neurite outgrowth inhibition caused by 4-sulfated CSPGs. (**a**) Schematic diagram showing actions of ARSB and ChABC on GAG chains. (**b**) Western blot showing CS-56 signal in conditioned medium. (**c**) Micrographs showing hippocampal neurons treated with no treatment, CSPG (5 μg/ml), or CSPG +ARSB. Scale bar = 25 μm. (**d**) Plot showing lengths of longest neurite measured from β-III-tubulin stained neurons. Statistical significance was determined by one-way ANOVA, **p<0.005, ***p<0.001. (**e**) Micrographs showing co-cultures of CGNs grown on astrocytes and treated with TGF-β, TGF-β and ARSB, or no treatment. Scale bar = 25 μm. (**f**) Plot showing lengths of longest neurite measured from β-III-tubulin stained neurons. Statistical significance was determined by one-way ANOVA, *p<0.05, **p<0.005, ***p<0.001, ****p<0.0001.

DOI: https://doi.org/10.7554/eLife.37139.002

The following figure supplement is available for figure 1:

**Figure supplement 1.** ARSB reverses neurite outgrowth inhibition caused by 4-sulfated CSPGs.

DOI: https://doi.org/10.7554/eLife.37139.003

To further validate that ARSB does not interfere with CSPG secretion, the level of CSPGs in CM was measured by immunoblotting with the antibody CS-56, which reacts with 4S and 6S groups on GAG chains (*Avnur and Geiger, 1984*). The increase in CSPGs caused by TGF-β treatment (*Wang et al., 2008*) was not altered by treatment with ARSB, even after repeated additions (*Figure 1b*), indicating that its enhancement of neurite growth was derived from modifying the sulfation pattern rather than attenuating CSPG production or secretion. These data also demonstrate that CS-56 immunoreactivity is not altered by removal of 4S from the non-reducing end of CS GAG chains.

## Optic nerve crush leads to astrogliosis and sustained elevation of CSPGs

Injury to the CNS is accompanied by astrogliosis, characterized by the accumulation of CSPGs in a glial scar (*Burnside and Bradbury, 2014*). Evidence suggests a similar effect in the optic nerve (*Brown et al., 2012*; *Sellés-Navarro et al., 2001*; *Sengottuvel et al., 2011*; *Qu and Jakobs, 2013*), but a comprehensive examination of this phenomenon has not been performed, especially regarding the production of 4S GAG chains following injury. A cohort of adult mice received optic nerve crush (ONC) or non-lesioned sham surgery (in which the nerve was exposed but not crushed) and optic nerves were collected at 1, 3, 5, 7, 14, and 21 days post crush (dpc). Optic nerve sections were stained with antibodies against GFAP (to identify reactive astrocytes) and Iba1 (to identify microglia and macrophages). By 7 dpc, GFAP$^+$ reactive astrocytes had retracted from the lesion site to form a cavity, which was densely populated with Iba1$^+$ microglia and macrophages (*Figure 2—figure supplement 1*). Astrocytes displayed reactive morphology, with elongated processes defining the lesion boundary. Some GFAP$^+$ cells were also found within the lesion core. Microglia in and around the lesion displayed an activated morphology, with enlarged cell bodies and retracted processes, distinct from the striated morphology of cells found distal to the injury site and in non-lesioned sham control nerves. By 21 dpc, astrocytes had begun to repopulate the cavity and form a chronic scar.

CSPGs were detected using CS-56 and 2H6, an antibody that reacts predominantly with 4S (*Yamamoto et al., 1995*), and to a lesser degree, with 6S (*Sugiura et al., 2012*) and 2,6S (*Matsushita et al., 2018*). In non-lesioned sham control nerves, CSPGs were evenly distributed within the tissue (*Figure 2a*). Elevated CS-56 levels at the lesion site were first observed at 5 dpc and peaked around 7 dpc (*Figure 2b*). Levels remained high at 21 dpc. An increase in 2H6 staining was also observed, with levels reaching 2.5-fold those in non-lesioned sham controls (fold change [mean ± SE]: 2.53 ± 0.15) (*Figure 2b*). The axons of injured mouse RGCs visualized with fluorescently-tagged CTB, injected intravitreally 1 d prior to tissue harvest, failed to traverse the injury site and instead formed dystrophic endbulbs that appeared to be associated with areas of high CSPG deposition, which included areas of high 4S immunostaining (*Figure 2c*). CSPGs and 4S GAGs were associated with both GFAP$^+$ and Iba1$^+$ cells in nerves examined at 7 dpc (*Figure 2—figure supplement 2*). Taken together, these results illustrate that ONC in mice leads to astrogliosis and elevated expression of CSPGs, especially those with 4S GAGs, which is sustained for at least 21 days.

## Modifying CSPG sulfation enhances retinal ganglion cell axon regeneration

Given the in vitro evidence that 4S is critical to CSPG-mediated inhibition of neurite growth, we investigated whether cleaving 4S from the non-reducing ends of GAG chains at the ONC lesion site would enhance retinal ganglion cell (RGC) axon regeneration in the optic nerve. To accomplish this, an intrinsic pro-regenerative stimulus, Zymosan A and CPT-cAMP (*Leon et al., 2000*; *Yin et al., 2003*), was combined with direct application of ARSB to the lesioned nerve. ChABC was used as a control to evaluate the effects of digesting GAG chains entirely rather than selectively removing 4S groups.

Mice received ONC, followed 3 days later by an intravitreal injection of Zymosan A (12.5 μg/μL) supplemented with CPT-cAMP (50 mM), followed immediately by implantation of a gelfoam scaffold loaded with 5 μL of ARSB (1 mg/mL), ChABC (455 μg/mL), or control buffer. At 14 dpc, optic nerves were dissected, sectioned, and stained for GAP-43 to detect regenerating axons. In accordance with previous reports (*Leaver et al., 2006*), we found that GAP-43 selectively labels regenerating axons, as GAP-43 signal is absent from intact, non-lesioned optic nerves (data not shown). On its own,

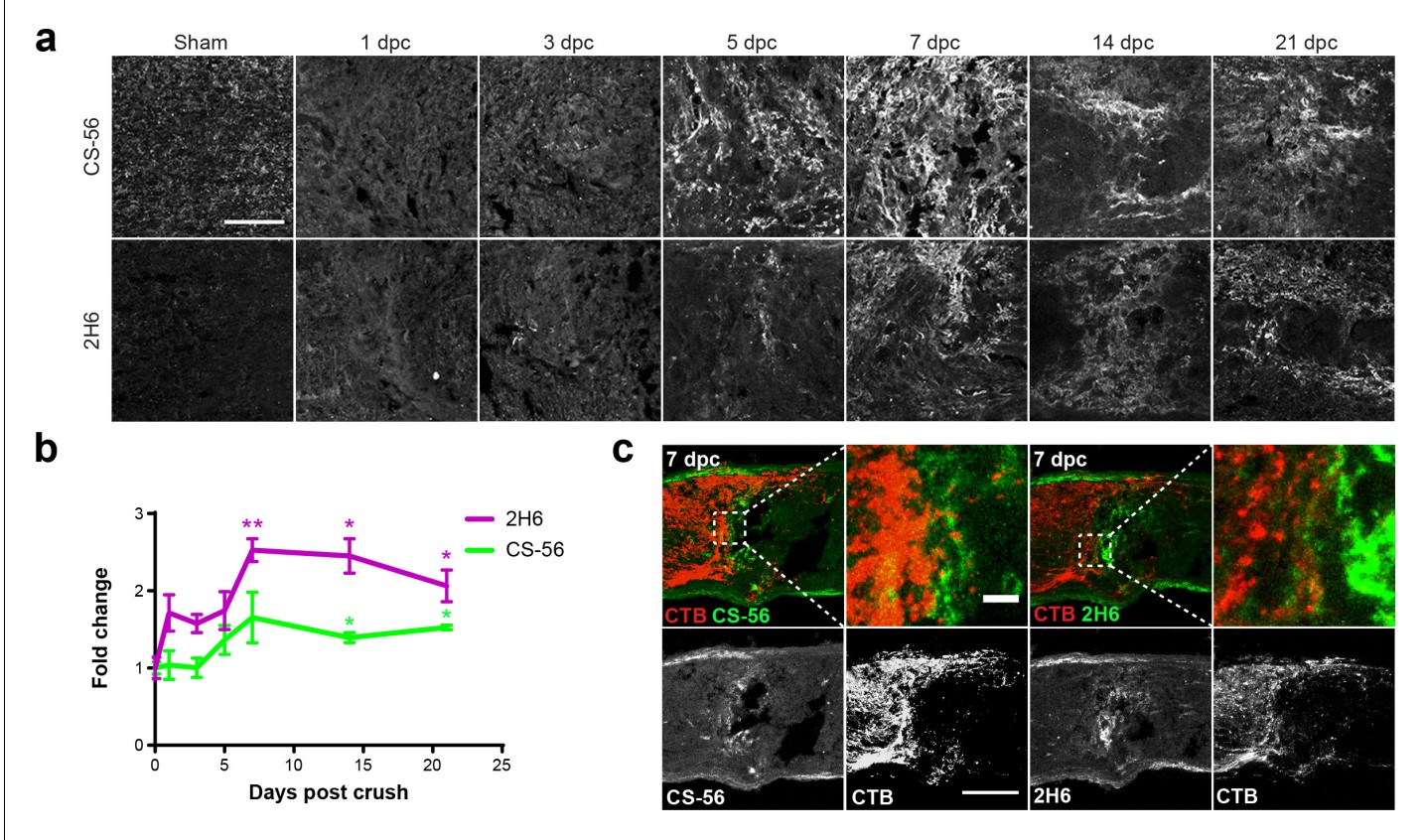

**Figure 2.** Optic nerve crush stimulates glial scar formation and sustained elevation of chondroitin sulfate proteoglycans. (a) Micrographs show lesioned optic nerve tissue collected at 1, 3, 5, 7, 14 and 21 dpc and stained for CSPGs (CS-56), 4S GAGs (2H6), reactive astrocytes (GFAP), and microglia and macrophages (Iba1). Scale bar = 50 µm. (b) Fluorescence intensity of CS-56 and 2H6 immunostaining expressed as fold change vs. non-lesioned sham controls. Statistical significance versus sham was determined by Student's t-test. *p<0.05, **p<0.005. Colored asterisks indicate significance for different groups (CS-56 = green, 2H6 = magenta). (c) Micrographs showing lesioned mouse optic nerve tissue at 7 dpc. Axons are visualized with CTB and form dystrophic endbulbs in areas of high CSPG and 4S GAG immunoreactivity. Scale bar = 100 µm, inset = 10 µm.

DOI: https://doi.org/10.7554/eLife.37139.004

The following figure supplements are available for figure 2:

**Figure supplement 1.** Reactive astrocytes and activated microglia form glial scar after optic nerve crush.

DOI: https://doi.org/10.7554/eLife.37139.005

**Figure supplement 2.** Optic nerve CSPGs associate with glial cells.

DOI: https://doi.org/10.7554/eLife.37139.006

injection of Zymosan/CPT-cAMP induced significantly (p=0.0226) higher RGC axon regeneration than PBS controls at 14 dpc (axons at 0.25 mm distal to the lesion [mean ±SE]: 282 ± 83.4 and 42.3 ± 11.1, respectively) (*Figure 3—figure supplement 1*). Zymosan did not alter CSPG expression at the lesion site (*Figure 3—figure supplement 1*). When Zymosan was combined with enzyme delivery, both ARSB and ChABC significantly (p=0.0006 and p<0.0001, respectively) enhanced RGC axon regeneration compared with the buffer control (axons at 0.25 mm distal to the lesion [mean ±SE]: 472 ± 62, 535 ± 123, and 217 ± 53, respectively) (*Figure 3*). Interestingly, delivering ARSB or ChABC in the absence of Zymosan injection did not enhance basal RGC axon regeneration (*Figure 3—figure supplement 2*).

The products of the reaction catalyzed by ARSB are not readily detectable by immunohistochemistry or Western blot; therefore, to specifically validate the penetration of ARSB into the optic nerve fibers, mice received ONC surgery, and gelfoam scaffolds soaked in 200 µg/mL His-Tagged ARSB or control buffer were implanted behind the eyes at the ONC lesion site (*Figure 3—figure supplement 3*). Tissue collected at 1 dpc was analyzed by immunohistochemistry using anti-His antibody, and recovered scaffolds were tested for the presence of active ARSB. His-Tagged ARSB was detected in

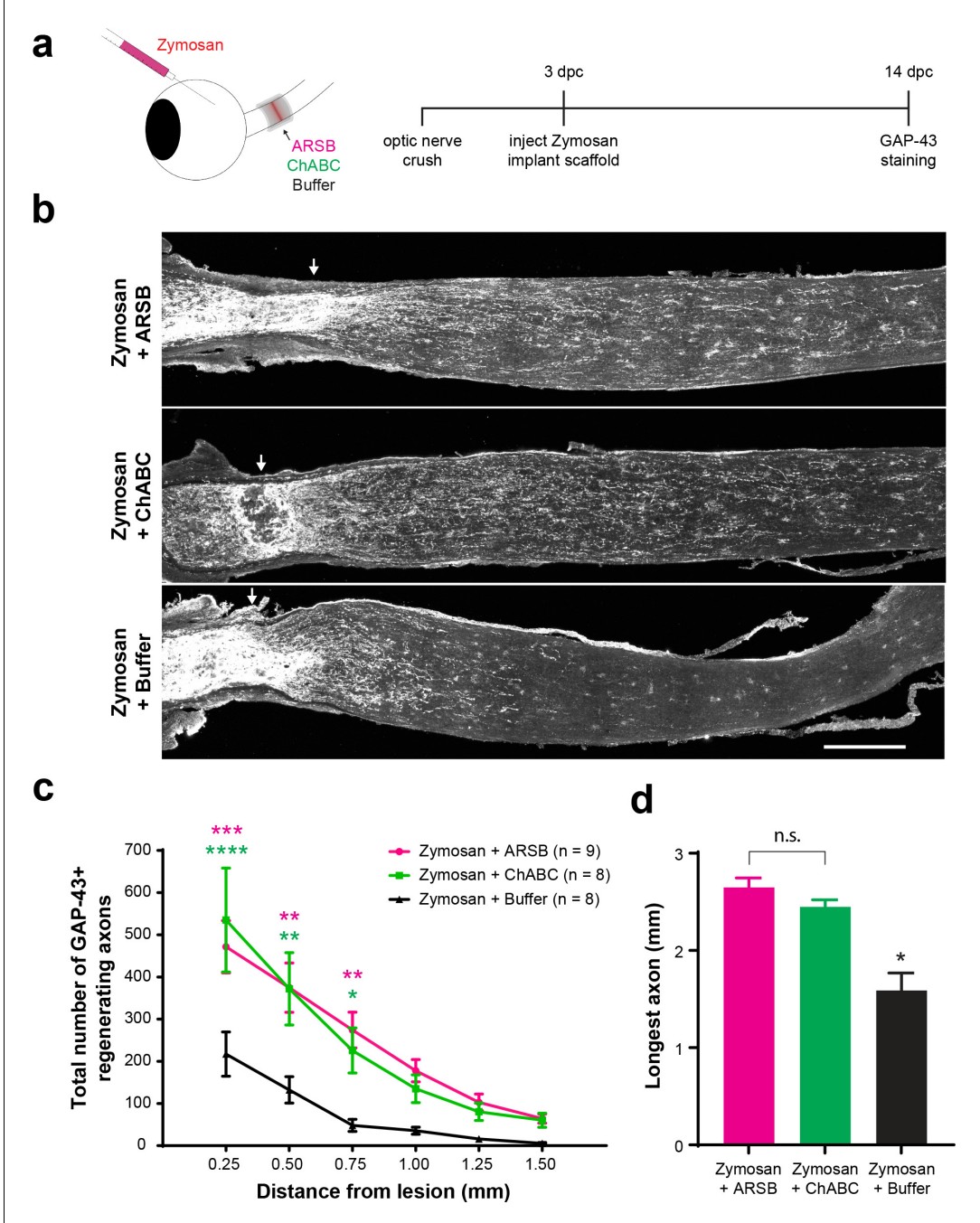

**Figure 3.** Selectively targeting inhibitory CSPGs enhances retinal ganglion cell axon regeneration. (a) Experiment timeline and schematic diagram showing intravitreal injection of Zymosan/CPT-cAMP and implantation of gelfoam scaffolds containing ARSB, ChABC, or control buffer. (b) Micrographs showing GAP-43-labeled optic nerves from mice treated with Zymosan/CPT-cAMP and gelfoam scaffolds loaded with ARSB, ChABC, or control buffer. Arrows indicate lesion site. Scale bar = 200 μm. (c) Graph showing the number of regenerating axons at distances distal to the lesion site, displayed as mean ± SEM. Statistical significance was determined by two-way ANOVA with Bonferroni post-hoc test for multiple comparisons. *p<0.05, **p<0.005, ***p<0.001, ****p<0.0001. Colored asterisks indicate statistical significance for different groups (ARSB = magenta, ChABC = green). (d) Graph showing average length of longest GAP-43+ regenerating axon. Statistical significance was determined by Student's t-test. *p<0.05.

DOI: https://doi.org/10.7554/eLife.37139.007

The following figure supplements are available for figure 3:

**Figure supplement 1.** Zymosan and CPT-cAMP stimulate axon regeneration.

DOI: https://doi.org/10.7554/eLife.37139.008

**Figure supplement 2.** CSPG-targeting enzymes alone do not induce axon regeneration.

*Figure 3 continued on next page*

*Figure 3 continued*

DOI: https://doi.org/10.7554/eLife.37139.009
**Figure supplement 3.** Tagged ARSB penetrates the optic nerve.
DOI: https://doi.org/10.7554/eLife.37139.010
**Figure supplement 4.** Implanted enzymes penetrate the optic nerve and modify GAG chains.
DOI: https://doi.org/10.7554/eLife.37139.011
**Figure supplement 5.** ARSB does not alter CS-56 or 2H6 immunoreactivity.
DOI: https://doi.org/10.7554/eLife.37139.012

lesioned tissue using immunohistochemistry, and active enzyme was detected from recovered scaffolds (*Figure 3—figure supplement 3*). To further validate that the enzymes had successfully penetrated the optic nerve and modified CSPGs, we stained ChABC-treated samples with the antibody BE-123, which recognizes the 'stubs' produced on proteoglycans by ChABC digestion of the GAG chains. Western blot analysis of non-lesioned sham control tissue treated with ChABC revealed BE-123 signal exclusively in nerve segments exposed to ChABC-loaded scaffolds (*Figure 3—figure supplement 4*). ARSB treatment did not significantly alter immunoreactivity of CS-56 or 2H6 (*Figure 3—figure supplement 5*). Together, these observations establish that the enzymes released from the scaffold penetrate the tissue and digest GAG chains.

## ARSB promotes axon regeneration with an extended therapeutic window

The duration of the regeneration enhancing effects of ARSB was assessed by measuring axon regeneration at early and late time points. At 7 dpc, only 4 days after implantation of the gelfoam scaffolds, a small but significant (p=0.0149) increase in the number of axons navigating through the lesion site was already detectable in the ARSB-treated group compared with the buffer control (axons at 0.50 mm distal to the lesion [mean ± SE]: 69.2 ± 12.3 and 16.0 ± 8.9, respectively) (*Figure 4a–d*). By 28 dpc, regenerating axons were found extending as far as 4.0 mm beyond the lesion site, to the optic chiasm entry point (*Figure 4e–g*). There was a significant (p=0.0002) increase in the number of axons in ARSB-treated animals versus buffer-treated controls (axons at 0.25 mm distal to the lesion [mean ± SE]: 568 ± 96.3 and 273 ± 63.0, respectively). The enhancing effect of ARSB treatment appeared to be concentrated at distances proximal to the lesion site (0.25–1.50 mm). At distances beyond 1.50 mm, there was relatively little difference between the ARSB-treated and buffer-treated groups (*Figure 4—figure supplement 1*). We isolated this effect by subtracting the number of regenerating axons in the Zymosan/buffer groups from those in the Zymosan/ARSB groups (*Figure 4—figure supplement 1*). ARSB strongly increased the number of axons regenerating through the lesion site but did not appear to substantially extend the distances of axons that were already regenerating.

## ARSB does not alter the astrocytic scar or perineuronal nets

To determine whether treatment with ARSB alters glial cells at the lesion site, tissue from enzyme-treated nerves was stained with GFAP and Iba1. Neither ARSB nor ChABC disrupted formation of the astrocytic scar. The area delineated by GFAP$^+$ astrocytes decreased over time but was not significantly different between treatment groups at any time point (*Figure 5c–d*). Correspondingly, the total GFAP immunoreactivity increased from 7 to 28 dpc as astrocytes repopulated the glial scar region, but no differences were observed between treatment groups (*Figure 5e*). Both ChABC and ARSB increased Iba1 immunoreactivity relative to the buffer control (fluorescence intensity [mean ± SE]: 21.7 ± 2.95, 12.9 ± 1.71, and 6.96 ± 1.79, respectively), but ChABC elicited significantly (p<0.05) higher Iba1 immunoreactivity than ARSB (*Figure 5a–b*).

In addition to their deposition in the glial scar, CSPGs are a major component of perineuronal nets (PNNs), structures that limit synaptic plasticity in the brain and spinal cord but are not present in the optic nerve. ChABC is known to disturb PNNs and alter plasticity in the visual cortex (*Pizzorusso et al., 2002*). To evaluate whether ARSB alters CSPG structure beyond the selective cleavage of 4S groups, we incubated post-fixed mouse brain tissue sections with ARSB (1 mg/mL), ChABC (≥20 µg/mL), and buffer control, and detected perineuronal nets (PNNs) with *Wisteria floribunda* agglutinin (WFA). ChABC completely eliminated WFA-stained PNNs (*Figure 5—figure*

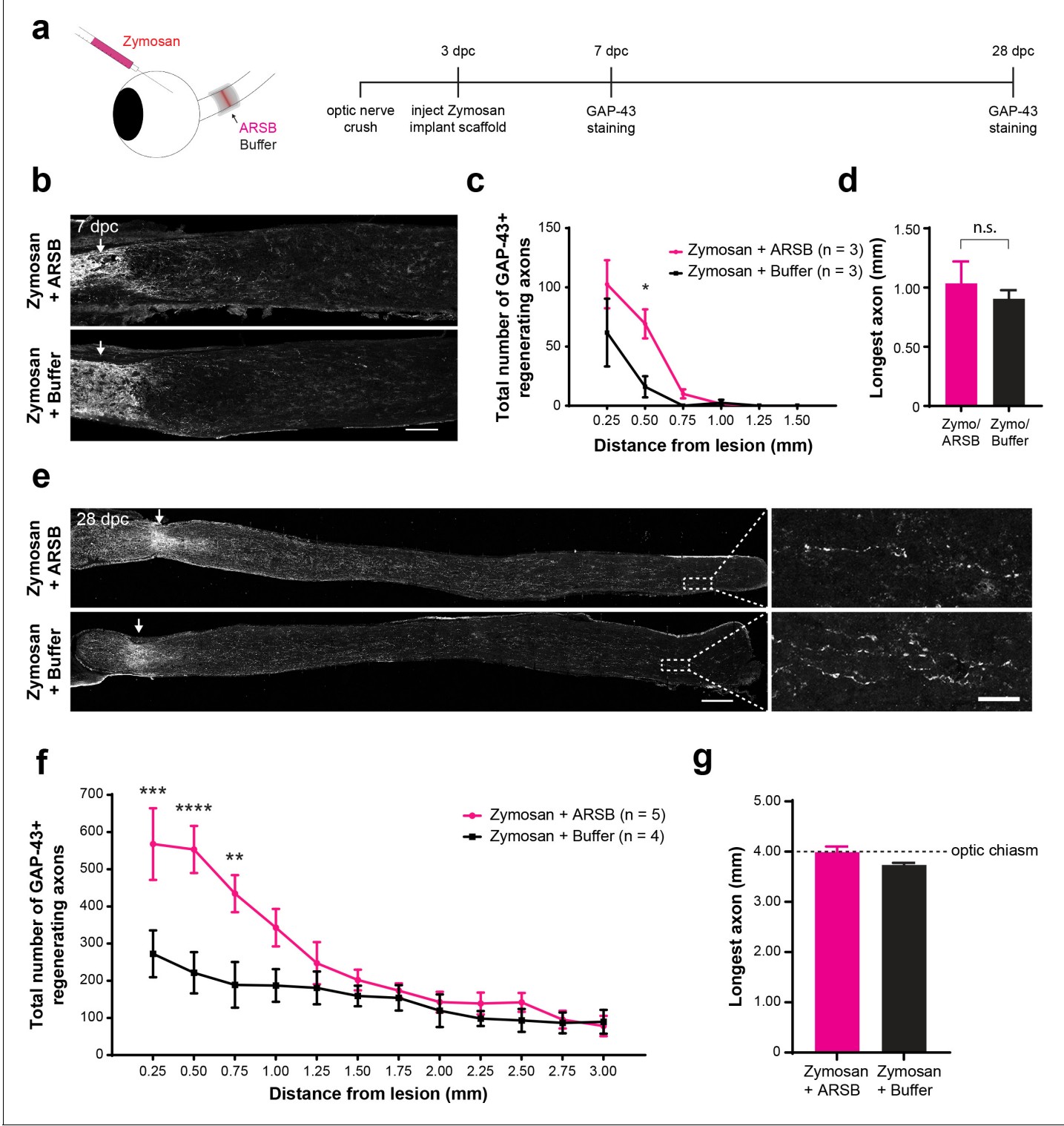

**Figure 4.** ARSB enhances axon regeneration over an extended therapeutic window. (**a**) Experiment timeline and schematic diagram showing intravitreal injection of Zymosan and CPT-cAMP and delivery of ARSB and control buffer to the lesioned optic nerve via implanted gelfoam scaffold. (**b**) Micrographs showing GAP-43-labeled optic nerves from mice treated with Zymosan/CPT-cAMP and gelfoam scaffolds loaded with ARSB or a control buffer. Arrows indicate lesion site. Scale bar = 200 μm. (**c**) Graph showing the number of regenerating axons at distances distal to the lesion site, displayed as mean ± SEM. Statistical significance was determined by two-way ANOVA with Bonferroni post-hoc test for multiple comparisons. *p<0.05. (**d**) Graph showing length of the longest regenerating axon, displayed as mean ± SEM. Statistical significance was determined by Student's t-test. (**e**) Micrographs showing GAP-43-labeled optic nerves from mice treated with intravitreal injections of Zymosan and gelfoam scaffold loaded with ARSB or

*Figure 4 continued on next page*

*Figure 4 continued*

a control buffer. Arrows indicate lesion site. Scale bar = 200 µm. (**f**) Graph showing the number of regenerating axons at distances distal to the lesion site, displayed as mean ± SEM. Statistical significance was determined by two-way ANOVA with Bonferroni post-hoc test for multiple comparisons. **p<0.005, ***p<0.001, ****p<0.0001. (**g**) Graph showing length of the longest regenerating axon, displayed as mean ± SEM. Statistical significance was determined by Student's t-test.

DOI: https://doi.org/10.7554/eLife.37139.013

The following figure supplement is available for figure 4:

**Figure supplement 1.** ARSB strongly enhances axon regeneration proximal to the lesion site.

DOI: https://doi.org/10.7554/eLife.37139.014

*supplement 1*). However, incubation with ARSB left PNNs intact, with no observable differences from PNNs in buffer-treated brain tissue (*Figure 5—figure supplement 1*).

## Discussion

The glial scar is considered a major impediment to axonal regeneration. We show that the injured optic nerve develops a glial scar rich in CSPGs, including the axon-inhibiting 4S motif. The human enzyme ARSB selectively cleaves 4S groups from the non-reducing ends of GAG chains, reducing CSPG-mediated inhibition of neurite growth in vitro. We demonstrate that ARSB promotes neurite growth in culture without altering production or secretion of GAG chains. Furthermore, ARSB enhances the regeneration of RGC axons following optic nerve injury. The treatment is robustly effective even when administered 3 days after injury, an important consideration for translational therapies. Enhanced regeneration was evident as early as 7 days post ONC and remained significant at 28 days, illustrating an extended therapeutic window from a single treatment. ARSB is active in vivo, provokes less Iba1 immunoreactivity than ChABC, and preserves perineuronal structures that depend on intact GAG chains. Taken together, these findings demonstrate that the 4S motif at the non-reducing end of CS GAG chains plays a major role in mediating the inhibitory actions of CSPGs. Given the approval for ARSB as an enzyme replacement therapy in human patients, our evidence that ARSB enhances axon regeneration in the optic nerve means that future treatments could readily combine ARSB with clinically viable intrinsic approaches to achieve robust regeneration of damaged or degenerated axons in the CNS.

### Sulfation dictates the effects of CSPGs on axon growth

Studies that link CSPGs to the failure of axon regeneration overwhelmingly fail to distinguish between differentially sulfated GAG chains, often showing instead that digestion of GAG chains with ChABC enhances neurite growth in vitro and axon regeneration in vivo (*Bradbury and Carter, 2011*). The importance of sulfation in governing CSPG function has been demonstrated using sodium chlorate, which broadly eliminates GAG sulfation (*Smith-Thomas et al., 1995*). Recent studies have characterized the behaviors of specific sulfation motifs, showing that both 4S and 4,6S inhibit neurite growth while 6S is growth-permissive (*Wang et al., 2008*; *Brown et al., 2012*). An age-related increase in the ratio of 4S to 6S was linked to declines in plasticity and memory (*Foscarin et al., 2017*; *Miyata et al., 2012*), and removal of 4S with ARSB improved motor function following spinal cord injury (*Yoo et al., 2013*). Blocking 4,6S with a custom antibody enhanced regeneration of RGC axons after ONC (*Brown et al., 2012*), which raises the question of whether 4S and 4,6S function similarly to inhibit axonal growth, and whether ARSB might convert 4,6S motifs to 6S.

The precise mechanism of how ARSB modifies the inhibitory actions of GAG chains is unknown. ARSB did not reduce the total amount of sulfated GAG in the culture medium as detected by the anti-CS antibodies, suggesting that its effects are mediated by altering GAG chain sulfation. ARSB, a lysosomal enzyme, maintains its highest activity at acidic pH, raising the question of whether it can cleave sulfate groups from secreted CSPGs, or whether lysosomal uptake is required. We observed that ARSB cleaves 4S from extracellular GAG chains in culture medium, suggesting that its activity at neutral pH is sufficient to perform its sulfatase function. This was validated by our discovery that ARSB promotes regeneration of optic nerve axons when administered exogenously. The prominent actions of ARSB are more remarkable considering that the average length of neuronal GAG chains is

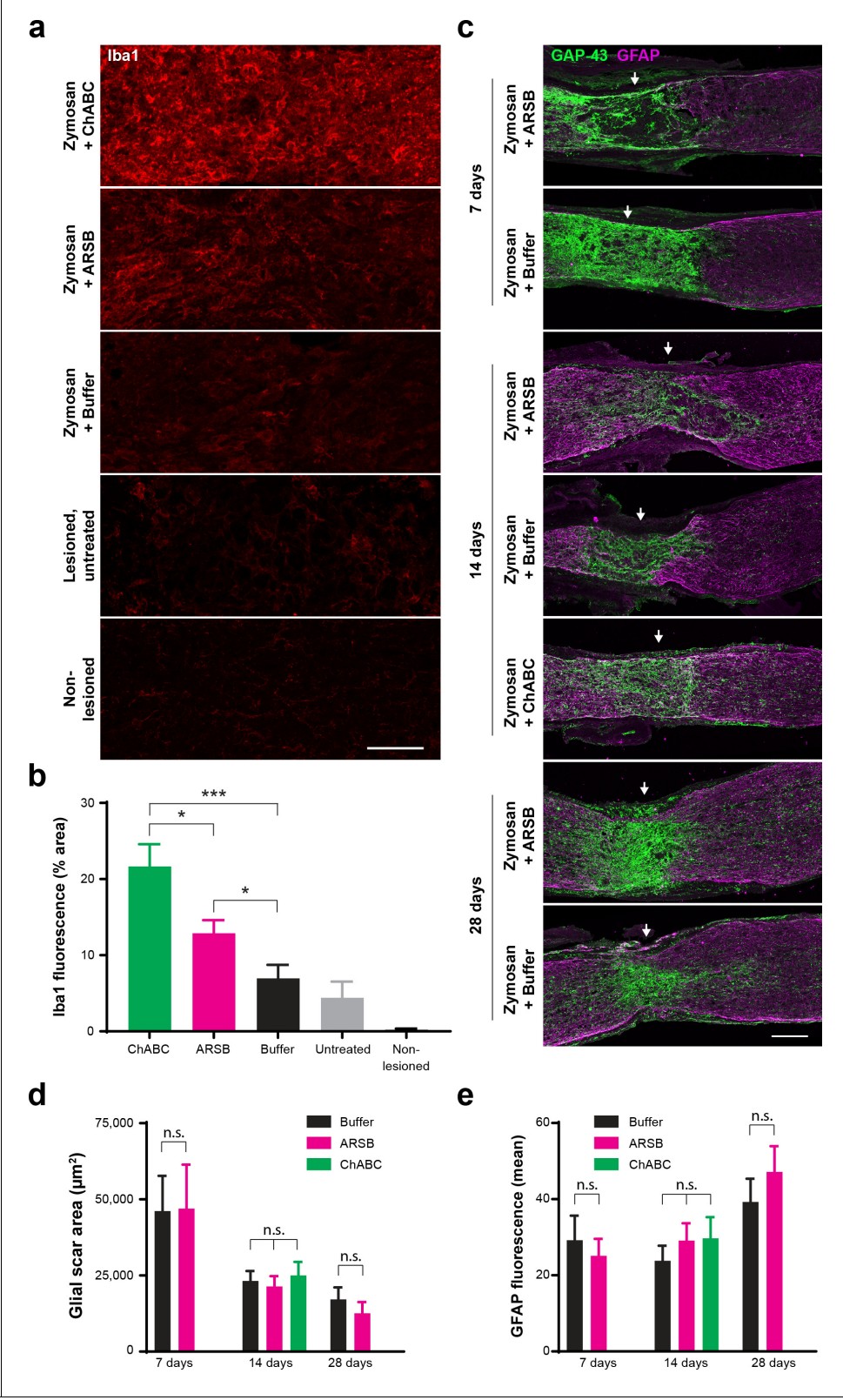

**Figure 5.** ARSB provokes muted immune response but does not alter astrocyte reactivity, glial scar size, or association of regenerating axons with astrocyte processes. (a) Micrographs showing Iba1 immunostaining at the optic nerve crush site for samples treated with Zymosan/CPT-cAMP and ChABC, Zymosan/CPT-cAMP and ARSB, Zymosan/CPT-cAMP and a control buffer, no treatment, and non-lesioned controls. Scale bar = 50 µm. (b) Graph showing quantification of Iba1 fluorescence intensity measured as % area of thresholded insets centered at the lesion site. Statistical significance

*Figure 5 continued*

was determined by Student's t-test. *p<0.05, ***p<0.001. (c) Micrographs showing GFAP and GAP-43 immunostaining at the optic nerve crush site for samples treated with Zymosan/CPT-cAMP and either ARSB, ChABC, or a control buffer and analyzed at 7, 14, and 28 dpc. Arrows indicate lesion site. Scale bar = 100 μm. (d) Graph showing quantification of glial scar size measured as the area delineated by GFAP⁺ astrocytes at the optic nerve crush site. Statistical significance was determined by Student's t-test. (e) Graph showing quantification of GFAP immunoreactivity at the optic nerve crush site. Statistical significance was determined by Student's t-test.

DOI: https://doi.org/10.7554/eLife.37139.015

The following figure supplement is available for figure 5:

**Figure supplement 1.** ARSB preserves CSPG-rich perineuronal net structure.

DOI: https://doi.org/10.7554/eLife.37139.016

about 50 disaccharide units (*Rauch et al., 1991*). Removal of just the 4S at the non-reducing end leaves virtually the entire GAG chain intact, as demonstrated by the preservation of the immunoreactivity to CS-56, while the inhibitory activity is significantly diminished.

## CSPG deposition is a key source of axon growth inhibition in the glial scar

The formation of a glial scar, including deposition of sulfated proteoglycans, is well documented in the brain and spinal cord (*Bradbury et al., 2002*; *Bradbury and Carter, 2011*; *Yi et al., 2012*; *Burnside and Bradbury, 2014*; *Galtrey and Fawcett, 2007*). Glial activation and macrophage recruitment have been observed in optic nerve lesions (*Qu and Jakobs, 2013*), and some studies have suggested that CSPGs are upregulated after ONC, but have not quantified this phenomenon or explored its time course (*Brown et al., 2012*; *Sengottuvel et al., 2011*; *Sellés-Navarro et al., 2001*). We found that CSPGs, and 4S GAGs in particular, were significantly elevated after ONC, reaching peak levels at 7 dpc. The sustained elevation of CSPGs at 21 dpc suggests that the optic nerve environment remains hostile to axon growth for extended periods after injury.

Cleaving 4S from the non-reducing ends of GAG chains with ARSB, or digesting GAG chains with ChABC, both enhanced axon regeneration without disrupting formation of the astrocytic scar. This supports a critical role for CSPG deposition, rather than scar formation per se, as the primary cause of axon growth inhibition. This is consistent with findings that ablating astrocytic scar formation without reducing CSPG levels does not lead to spontaneous regeneration of axons (*Anderson et al., 2016*; *Silver, 2016*). Intriguingly, blocking the transformation of reactive astrocytes into scar-forming astrocytes, which express elevated levels of CSPG-related transcripts, was found to significantly enhance axon regeneration (*Hara et al., 2017*).

## Delayed application of ARSB promotes regeneration and preserves perineuronal structures

Most experimental therapies that stimulate RGC axon regeneration involve interventions at the time of injury or, in the case of many gene therapies, prior to injury (*Buch et al., 2008*). While such studies are valuable for identifying therapeutic targets and elucidating mechanisms of RGC axon regeneration, they are not readily translatable to human patients. We found that delivery of ARSB in conjunction with Zymosan/CPT-cAMP significantly enhanced RGC axon regeneration when administered 3 days after ONC, making a strong case for its clinical viability.

Delaying ARSB treatment may confer additional advantages: previous studies have argued that CSPG synthesis in the acute phase (0–2 dpi) may, in fact, promote recovery. *Rolls et al., 2008* blocked CSPG synthesis immediately after spinal cord injury in mice and found that axon regeneration and functional recovery were impaired, whereas blocking synthesis in the subacute phase (2–7 dpi) enhanced regeneration and recovery. Other studies suggest that intact CSPGs recruit blood-borne monocytes and bias macrophages toward a resolving phenotype (*Shechter et al., 2011*), and that CSPGs regulate the spatial organization of microglia and macrophages and promote neurotrophic factor production by resident microglia (*Rolls et al., 2008*; *Shechter et al., 2009*). Stripping CSPGs of their GAG chains may impede these repair functions, whereas selectively modifying sulfation with ARSB could reduce GAG-mediated inhibition of neurons without disrupting their interactions with other cells. To demonstrate that ARSB preserves perineuronal structures composed of

CSPGs, we treated mouse cortical tissue with ARSB and ChABC and showed that while ChABC eliminated perineuronal nets (PNNs), ARSB left PNNs intact.

ARSB is superior to ChABC in several other respects. ARSB has relatively lower immunogenicity and maintains its activity longer than ChABC in vitro (*Yoo et al., 2013*). In vivo studies indicate that ChABC injected into rat brains maintains activity for at least 10 days (*Lin et al., 2008*), and that low levels of ChABC activity suppress CSPG levels for weeks (*Chau et al., 2004*; *Hyatt et al., 2010*). While the durability of ARSB in vivo has not been characterized, ARSB is stable at physiological temperature and pH, meaning it should retain robust levels of activity for extended periods (*Yoo et al., 2013*). Crucially, ARSB is a human enzyme with preexisting approval for clinical use (*Muñoz-Rojas et al., 2010*; *Harmatz et al., 2004*; *Harmatz et al., 2005*).

### Combining extrinsic and intrinsic stimuli enhances axon regeneration

Treating lesioned optic nerves with ARSB or ChABC alone failed to enhance regeneration, but combining them with Zymosan/CPT-cAMP promoted significantly greater regeneration than the intrinsic treatment alone. Most studies demonstrating long distance regeneration of RGC axons achieve their effects by modifying the intrinsic state of RGCs: knocking out the tumor suppressor PTEN (*Park et al., 2008*), delivering growth factors (*Sieving et al., 2006*), stimulating inflammatory pathways (*Yin et al., 2003*), enhancing the endogenous activity of RGCs (*Lim et al., 2016*), chelating neurotoxic ions in the retina (*Li et al., 2017*), and various combinations thereof. However, knowledge of how these regenerating axons traverse the glial scar and navigate the growth-inhibitory microenvironment is incomplete. Studies that have examined the three-dimensional growth patterns of regenerating RGC axons consistently find that axons induced to regenerate via intrinsic manipulations display highly irregular and aberrant growth patterns (*Luo et al., 2013*; *Bray et al., 2017*; *Fischer et al., 2017*). Understanding how axons respond to their extrinsic microenvironment, particularly GAG chains within the glial scar, will be vital to future efforts to stimulate robust long-distance regeneration of retinal neurons and successful innervation of visual targets in the brain.

## Materials and methods

**Key resources table**

| Reagent type (species) or resource | Designation | Source or reference | Identifiers | Additional information |
|---|---|---|---|---|
| Antibody | DAPI | ThermoFisher Scientific | D3751; RRID: AB_2307445 | 1/10,000 |
| Antibody | anti-GAP-43 (Sheep) | *Benowitz et al., 1988*; PMID: 3339416 | | 1/50,000; Gift from Benowitz lab |
| Antibody | anti-GAP43 (Rabbit, polyclonal) | Abcam | ab7462; RRID: AB_305932 | 1/500 |
| Antibody | CS-56 (Mouse, monoclonal) | Millipore Sigma | C8035; RRID: AB_476879 | 1/500 |
| Antibody | 2H6 (Mouse, monoclonal) | Amsbio | 370710-IEC | 1/500 |
| Antibody | BE-123 (Mouse, monoclonal) | Millipore Sigma | MAB2030; RRID: AB_94510 | 1/500 |
| Antibody | Iba1 (Rabbit, polyclonal) | FUJIFILM Wako Chemicals USA | 019–19741; RRID: AB_839504 | 1/500 |
| Antibody | GFAP (Rabbit, polyclonal) | Agilent (Dako) | Z0334; RRID: AB_10013382 | 1/500 |
| Antibody | GFAP (Chicken, polyclonal) | Abcam | ab74674; RRID: AB_304558 | 1/500 |
| Antibody | β-III-tubulin (Mouse, monoclonal) | Millipore Sigma | T8660; RRID: AB_477590 | 1/1000 |
| Antibody | 6x His tag | Abcam | ab137839 | 1/500 |

*Continued on next page*

*Continued*

| Reagent type (species) or resource | Designation | Source or reference | Identifiers | Additional information |
|---|---|---|---|---|
| Antibody | WFA | Millipore Sigma | L1516; RRID: AB_2620171 | 1/500 |
| Antibody | Donkey anti-sheep, Alexa 488 (secondary) | ThermoFisher Scientific | A-11015; RRID: AB_2534082 | 1/1000 |
| Antibody | Donkey anti-sheep, Alexa 568 (secondary) | ThermoFisher Scientific | A-21099; RRID: AB_2535753 | 1/1000 |
| Antibody | Goat anti-rabbit, Oregon Green 488 (secondary) | ThermoFisher Scientific | O-6381; RRID: AB_2539800 | 1/1000 |
| Antibody | Goat anti-rabbit, Alexa 633 (secondary) | ThermoFisher Scientific | A-21070; RRID: AB_2535731 | 1/1000 |
| Antibody | Goat anti-chicken, Alexa 488 (secondary) | ThermoFisher Scientific | A-11039; RRID: AB_2534096 | 1/1000 |
| Antibody | Goat anti-mouse, Alexa 568 (secondary) | ThermoFisher Scientific | A-11004; RRID: AB_2534072 | 1/1000 |
| Antibody | Goat anti-mouse IgM mu chain, Dylight 650 (secondary) | Abcam | ab98749; RRID: AB_10672799 | 1/500 |
| Antibody | TRITC-conjugated streptavidin (secondary) | Jackson ImmunoResearch | 016-020-08 | 1/1000 |
| Recombinant Protein | TGF-β (human) | PeproTech | 100–21C | |
| Recombinant Protein | ARSB (human) | BioMarin Pharmaceuticals | | Naglazyme, Provided by BioMarin |
| Recombinant Protein | ARSB (human) | R and D Systems | 4415-SU-010 | |
| Recombinant Protein | Cholera Toxin Subunit B, Alexa 555 | ThermoFisher Scientific | CC22843 | |
| Protein | ChABC | Millipore Sigma | C3667 | |
| Protein | CSPG (Chicken Extracellular Chondroitin Sulfate Proteoglycans) | Millipore Sigma | CC117 | |
| Chemical compound, drug | Zymosan A | Millipore Sigma | Z4250 | |
| Chemical compound, drug | CPT-cAMP | Millipore Sigma | C3912 | |
| Chemical compound, drug | PNCS | Alfa Aesar | B23325 | |
| Chemical compound, other | Can Get Signal (Immunoenhancer) | CosmoBio | TYB-NKB-101 | |
| Software, algorithm | GraphPad Prism 7 | | RRID:SCR_002798 | |
| Software, algorithm | Adobe Illustrator CC 2017 | | RRID:SCR_010279 | |
| Software, algorithm | Fiji | | RRID:SCR_002285 | |
| Other | Gelfoam | Ethicon- Johnson and Johnson | 1972 | |

## Animals

All experiments and procedures were performed in accordance with protocols approved by the Institutional Animal Care and Use Committee (IACUC) at the National Institutes of Health. Female

C57Bl/6 mice aged 6–8 weeks (Charles River) were housed in a pathogen free facility with free access to food and a standard 12 hr light/dark cycle. Sample sizes were determined by statistical power calculations from pilot experiments and the results of previous studies, as described below. Animals were randomly allocated into experimental groups. Animals were removed from the study if bleeding occurred during the optic nerve crush or scaffold implantation surgery.

## Cell culture

Primary hippocampal neuron cultures were prepared from embryonic (e17-18) C57Bl/6 mouse brains. Hippocampi were dissected and dissociated into single cell suspensions. Dissociated cells were seeded onto coverslips coated with poly-L-lysine and cultured in 500 µL Neurobasal medium containing B27 supplement (Thermo Fisher) and 24 mM KCl. After allowing 2 hr for neuronal attachment, 500 µL of Neurobasal medium containing B27 supplement and 24 mM KCl that had been incubated for 4 hr with no treatment, 10 µg/ml CSPG (for final concentration of 5 µg/ml), or CSPG (10 µg/ml)+ARSB (2 µg/ml) (final concentrations 5 µg/ml and 1 µg/ml, respectively) was added. Cells were incubated for 48 hr at 37°C and 5% $CO_2$ atmosphere and then fixed and stained for DAPI and βIII-tubulin.

Primary cortical astrocyte cultures were prepared from neonatal (1–3 days) C57Bl/6 mouse brains as described previously (*Wang et al., 2008*). Cerebral cortices were dissected and dissociated into single cell suspension. Dissociated cells were seeded into T-75 flasks and grown in Dulbecco's Modified Eagle's Medium (DMEM) supplemented with 10% fetal bovine serum (FBS) at 37°C and 5% $CO_2$ atmosphere until cells grew to confluence (10–14 days). Flasks were shaken for 20 hr (120 rpm, 37°C) to detach microglia, oligodendrocytes, and neurons from the more adherent astrocytes. After shaking, the medium was replaced. Media replacement was repeated 24 hr after the shaking period.

To harvest conditioned media from reactive astrocytes, purified astrocytes were plated into T-75 flasks in serum-containing medium. After reaching confluence, astrocytes were incubated with serum-free media overnight and treated with TGF-β (10 ng/mL), ARSB (1 µg/mL), TGF-β and ARSB, or neither (untreated controls), for 7 days. After harvesting, conditioned media was centrifuged at 800 rpm for 5 min to remove debris before being split into three aliquots of 2 mL each. Aliquots were treated with no enzyme, ARSB (1 µg/mL), or ChABC (1 µL/mL) for 4 hr prior to addition to neuronal cultures.

Cerebellar granule neurons (CGNs) were isolated at previously described (*Wang et al., 2008*). Dissociated cells were cultured in 500 µL Neurobasal medium containing B27 supplement and 24 mM KCl and plated on poly-L-lysine-coated coverslips in 24-well plates. After allowing 2 hr for neuronal attachment, 500 µL of treated conditioned medium was applied to each well in triplicate. Cells were incubated for 24 hr and then fixed and stained for DAPI and βIII-tubulin. In co-culture experiments, dissociated CGNs were plated at a density of $5 \times 10^4$ cells/well onto a confluent monolayer of astrocytes in 24-well plates that had been treated for 7 d with ARSB (1 µg/mL), TGF-β (10 ng/mL) or TGF-β and ARSB.

## Neurite outgrowth analysis

After fixation and staining, at least 60 images were taken across two coverslips per condition. Files were analyzed by an experimenter blinded to the experimental conditions. Neurons were measured if they were isolated from other neurons and had distinct nuclei and at least one neurite longer than the diameter of the cell body. The longest neurite was measured for each neuron and at least 60 neurons were measured for each condition. Each experiment was performed in triplicate.

## Preparation of zymosan/CPT-cAMP and enzyme delivery scaffolds

In accordance with established protocols (*Yin et al., 2003*; *de Lima et al., 2012*), Zymosan A (Sigma Z4250) was suspended in sterile PBS at a concentration of 12.5 µg/µL, incubated at 37°C for 10 min, and vortexed. CPT-cAMP (Sigma C3912) was added to achieve a final concentration of 50 mM CPT-cAMP. Aliquots were stored at 4°C for up to two weeks. ChABC (Amsbio 100332-1A) was reconstituted at 455 µg/mL in a buffer solution containing 100 mM Tris(hydroxymethyl)aminomethane hydrochloride (Tris-HCl) and 0.1% BSA in 1X phosphate buffered saline (PBS). ARSB (Naglazyme) was obtained in acidic PBS (pH 5.5) from Biomarin (San Rafael, CA). ARSB with His Tag was obtained from R and D Systems (4415-SU). Sterile gelfoam sponges were cut to roughly 2 $mm^3$ and placed to

soak in a sterile tube containing 5 µL of either ChABC, ARSB, or the control buffer. Tubes were stored on ice for up to 4 hr before surgical implantation.

## Optic nerve crush and implantation of enzyme scaffolds

Optic nerve crush was performed as described previously (*Park et al., 2008*). Mice were anesthetized using 1–2% isoflurane; the depth of anesthesia was confirmed by lack of response to a toe pinch. The optic nerve was exposed intraorbitally, and curved forceps were inserted beneath the external ocular muscle, avoiding the ophthalmic artery and retrobulbar sinus. The nerve was crushed approximately 1 mm behind the eye for 10 s. Immediately after the crush, eyes were monitored fundoscopically for signs of ischemia, and mice were observed for bleeding in the hours following surgery. Mice received a subcutaneous injection of 1 mg/kg buprenorphrine as an analgesic and topical application of ophthalmic ointment to prevent corneal drying.

For implantation of enzyme scaffolds, the optic nerve was exposed by gently reopening the conjunctiva and inserting curved forceps behind the eye. Carefully avoiding the ophthalmic artery and retrobulbar sinus, the enzyme- or buffer-soaked gelfoam scaffold was placed in direct contact with the optic nerve at the site of the crush lesion, approximately 1 mm behind the eye. Retinal blood flow was assessed fundoscopically, and mice received a subcutaneous injection of 1 mg/kg buprenorphrine and topical application of ophthalmic ointment.

## Intravitreal injection

Intravitreal injections of Zymosan or a PBS control were administered immediately following implantation of the gelfoam scaffold, and injections of CTB were administered 1 d prior to tissue harvest. 2 µL of the injecting solution was drawn into a sterile 5 µL Hamilton syringe with a 33-gauge removable needle. In the case of Zymosan injections, the syringe was inspected to ensure that the needle was not blocked by Zymosan particles. The solution was then slowly injected through the superior nasal sclera at a 45° angle, avoiding the lens, external ocular muscle, and blood vessels. A sterile 33-gauge needle was used to puncture the cornea and drain the anterior chamber before removing the injecting needle, to reduce intraocular pressure and prevent reflux of the injected solution. Different needles were used for Zymosan and PBS injections to prevent contamination, and the syringe was rinsed thoroughly with ethanol followed by sterile PBS between injections.

## Western blot
### Tissue preparation
Mice were anesthetized using 1–2% isoflurane and exsanguinated, followed by cervical dislocation. Optic nerves were severed between the globe and the optic and cut into four equally sized segments of approximately 1.0–1.5 mm each. Nerve segments were immediately placed in sterile 1.5 mL Eppendorf tubes containing cold 40 µL lysis buffer (cOmplete Lysis-M, EDTA-free, Roche). Tissue was mechanically homogenized using a sterile pestle and centrifuged to separate un-homogenized tissue. Protein concentration in the supernatant was determined using the BCA assay (Thermo-Fisher). Samples were frozen and stored at −80°C.

### Immunoblotting
Proteins were separated by SDS-PAGE under reducing conditions and transferred to a 0.45 µm PVDF membrane. Membranes were blocked with PBS containing 0.2% Tween-20% and 5% skim milk for 1 hr at room temperature. To detect ChABC-digested CSPGs, membranes were incubated with the primary mouse monoclonal antibody BE-123 (Millipore MAB2030) diluted in an immunoenhancing reagent (Can Get Signal, Toyobo) and 5% skim milk for 2 hr at 4°C, then washed and incubated with an HRP-conjugated anti-mouse IgG secondary antibody for 30 min at room temperature. Signals were visualized with myECL[TH] Imager (ThermoFisher, Waltham, MA, USA).

## Enzyme activity
Activity of ChABC and ARSB was assessed immediately before surgery. ChABC activity was measured by spectrophotometrically detecting the production of disaccharides cleaved from the glycosaminoglycan chains of CSPGs, as has been previously described (*Suzuki et al., 1968*).

ARSB activity was measured by detecting the cleavage of a sulfate group from p-nitrocatechol sulfate (PNCS), which yields a product with an absorbance peak at 510 nm (*Porter et al., 1969*; *Knaust et al., 1998*). To measure enzyme activity after in vivo implantation, scaffolds were recovered from freshly dissected optic nerves, placed in 1.5 mL Eppendorf tubes, and stored on ice. 250 µL of 50 mM 2-(N-morpholino)ethanesulfonic acid (MES) at pH 6.5 was added to the tube containing the recovered scaffold. After approximately 1 hr, three aliquots of 75 µL were removed from this solution and combined with 75 µL 4-PNCS in a 96-well microplate. Samples were incubated at 37°C for 24 hr, after which the reaction was quenched by adding 150 µL of 0.2 N NaOH. Absorbance was measured at 510 nm. Recovered scaffolds loaded with enzyme buffer served as controls.

## Enzyme treatment of brain sections

Free-floating 30 µm sections of mouse brain were incubated with either ChABC (Sigma C3667,$\geq$20 µg/mL), ARSB (Naglazyme, Biomarin, 1 mg/mL), or control buffer (50 mM Tris, 60 mM sodium acetate, and 0.02% BSA, pH 8.0) in a 24-well plate. ChABC and ARSB were assayed to confirm activity before being added to the wells. Brain sections were incubated with enzyme and control solutions at 37°C for 8 hr.

## Immunohistochemistry

### Tissue preparation

Mice were anesthetized using 1–2% isoflurane and transcardially perfused with PBS followed by 4% paraformaldehyde (PFA). Optic nerves were dissected, laid flat on 13 mm filter paper (Millipore AABG01300), and immersed in 4% PFA. The tissue was post-fixed overnight, then immersed in 30% sucrose for at least 24 hr for cryoprotection. Nerves were embedded in Tissue-Tek OCT and snap-frozen for cryosectioning. 14 µm longitudinal sections were obtained on charged Superfrost microscope slides using a Leica CM3050 cryostat. Slides were dried and stored at −80°C.

For analysis of perineuronal nets, fresh brain tissue was dissected from a C57Bl/6 mouse and immediately immersed in 4% PFA. Tissue was post-fixed for 24 hr, cryoprotected in 30% sucrose for 24 hr, embedded in Tissue-Tek OCT, and snap-frozen for sectioning.

### Immunostaining

For antibodies detecting CSPGs and glial cell activation, slides with optic nerve sections were incubated for 1 hr in blocking solution (PBS containing 3% goat serum and 0.2% Triton X-100), then incubated overnight at 4°C in primary antibodies diluted in the blocking solution. Slides were washed three times for 5 min with PBS, incubated for 2 hr with secondary antibodies, washed, and mounted onto glass coverslips with Fluoromount medium (Sigma).

The GAP-43 antibody (*Benowitz et al., 1988*) was incubated as previously described (*Leon et al., 2000*). Briefly, slides were rinsed in TBS (50 mM Tris buffer containing 8.766 g/L NaCl) and then washed with methanol for 10 min. Slides were blocked in TBS containing 10% donkey serum for 1 hr. The GAP-43 antibody was diluted 1:50,000 in a solution of TBS$_2$T (50 mM Tris buffer, 17.532 g/L NaCl, and 0.1% Tween) containing 5% donkey serum and 2% BSA. Slides were incubated with primary antibody overnight on a rocking platform. Slides were then washed with TBS$_2$T for 1 hr, with TBS$_2$T plus 5% donkey serum and 2% BSA for 1 hr, and with TBS$_2$T for 1 hr, all on a rocking platform. The secondary antibody was diluted 1:1000 in TBS$_2$T plus 5% donkey serum and 2% BSA. Slides were incubated with the secondary antibody solution for 2 hr, followed by 30 min washes with TBS$_2$T, TBS$_2$T, and TBS. Slides were mounted using Fluormount and glass cover slips, and stored at 4°C for imaging.

For detection of perineuronal nets, immediately after incubation with enzymes, free-floating brain sections were washed with 1 mL of PBS containing 0.02% Triton-X100 three times for 30 min. Sections were incubated with 250 µL biotinylated *Wisteria floribunda* agglutinin (WFA) overnight at 4°C on a rocking platform. Sections were then washed with 1 mL PBS/0.02% Triton three times for 5 min, incubated with 250 µL TRITC-conjugated streptavidin for 1 hr at room temperature, washed with 1 mL PBS three times for 5 min, stained with DAPI, and mounted using Fluoromount and glass cover slips. Slides were stored at 4°C prior to imaging.

## Microscopy and Image Processing

Tissue was imaged using a Zeiss 780 confocal microscope with 40X and 63X objectives. Z-stacks were maximally projected onto a single plane using Zeiss image processing software. For images used in fluorescence quantification, image capture settings were held constant, and samples from within each group were imaged at the same time. Fluorescence intensity was measured using ImageJ, with identical settings for all samples within each analysis.

### Quantification of regenerating axons

In ImageJ, vertical lines were drawn through each nerve section at 0.25 mm intervals starting from the lesion site, and the number of GAP-43$^+$ axons crossing each line was manually counted. Four sections were counted for each nerve. The number of regenerating axons per nerve was then calculated at each distance using a previously developed formula (*Lim et al., 2016*; *Bei et al., 2016*), with the total number of axons equal to $\pi r^2$ (r being the maximum recorded radius of the optic nerve section) times the average number of counted axons, divided by the thickness of the section (14 μm). Axon counting was verified by a separate observer blind to the experimental conditions. For quantification of longest axon, the same images were used. GAP-43$^+$ axons were identified, and the length of the longest detectable axon was measured from the lesion site using ImageJ.

### Statistics

Sample size for axon regeneration experiments was determined based on preliminary data from a pilot experiment. The number of regenerating axons counted at 0.50 mm distal from the lesion site was obtained from groups of mice treated with either Zymosan + ARSB (n = 4) or Zymosan + Buffer (n = 5). The control group had a mean of 104 ± 53 axons at 0.50 mm, while the ARSB-treated group had a mean of 260 ± 84 axons. Based on these numbers, we assumed a standard deviation of 75, to be equal for each group, and estimated using a two-sided two sample *t*-test that n = 9 mice per group would be required to achieve 80% power (at the 0.025 level) to compare ARSB treatment to a buffer control.

All statistical tests were performed using GraphPad Prism 7.0 (GraphPad Software, La Jolla, CA). Neurite lengths in culture were compared using Kruskal-Wallis Analysis of Variance and Mann-Whitney U tests. Axon regeneration was assessed using two-way ANOVA and Bonferroni post-hoc analysis. Asterisks indicate significance levels as specified in the corresponding figure legends.

## Acknowledgements

This research was supported by the Division of Intramural Research of the National Heart, Lung, and Blood Institute, the Cambridge Eye Trust, Fight for Sight UK, and the Jukes Glaucoma Research Fund. Help with statistical analyses was provided by Dr. Nancy Geller. We also appreciate the input and suggestions of Dr. Yasuhiro Katagiri. GAP-43 antibody was generously provided by Drs. Larry Benowitz and Yuqin Yin.

## Additional information

### Funding

| Funder | Grant reference number | Author |
| --- | --- | --- |
| National Institutes of Health | 1ZIAHL006135 | Craig S Pearson<br>Caitlin P Mencio<br>Herbert M Geller |
| Cambridge Eye Trust | | Amanda C Barber<br>Keith R Martin |

The funders had no role in study design, data collection and interpretation, or the decision to submit the work for publication.

## Author contributions
Craig S Pearson, Conceptualization, Data curation, Formal analysis, Investigation, Writing—original draft; Caitlin P Mencio, Conceptualization, Data curation, Formal analysis, Investigation, Methodology, Writing—review and editing; Amanda C Barber, Conceptualization, Writing—review and editing; Keith R Martin, Conceptualization, Resources, Supervision, Funding acquisition, Writing—review and editing; Herbert M Geller, Conceptualization, Resources, Data curation, Supervision, Funding acquisition, Methodology, Project administration, Writing—review and editing

## Author ORCIDs
Craig S Pearson http://orcid.org/0000-0002-1906-6347
Herbert M Geller http://orcid.org/0000-0002-7048-6144

## Ethics
Animal experimentation: This study was performed in strict accordance with the recommendations in the Guide for the Care and Use of Laboratory Animals of the National Institutes of Health. All of the animals were handled according to approved institutional animal care and use committee (IACUC) protocols (#H-0186R3) of the National Heart, Lung, and Blood Institute, NIH. Mice were anesthetized using 1-2% isoflurane, and every effort was made to minimize suffering.

## Decision letter and Author response
Decision letter https://doi.org/10.7554/eLife.37139.019
Author response https://doi.org/10.7554/eLife.37139.020

## Additional files

### Supplementary files
• Transparent reporting form
DOI: https://doi.org/10.7554/eLife.37139.017

### Data availability
All data generated or analysed during this study are included in the manuscript and supporting files.

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
