## [Decision Letter]

Thank you for submitting your work entitled "Identification of a critical sulfation in chondroitin that inhibits axonal regeneration" for consideration by *eLife*. Your article has been reviewed by three peer reviewers, and the evaluation has been overseen by a Reviewing Editor and a Senior Editor. The following individuals involved in review of your submission have agreed to reveal their identity: Hudson H Freeze (Reviewer #3).

Our decision has been reached after consultation between the reviewers. All three reviewers appreciate the quality of the science and the potential impact of chondroitin sulfating in axon regeneration. They appreciated the quality of the cell culture and animal model data, as well as the potential tor translatability to axon regeneration. However, they identified a number of flaws that will preclude publication in its current form. These center around the choice of the experimental system, the need to prime the system with zymosan, and size of the effect observed, and how these results are quantitated. Addressing these issues may be within the scope of what is possible, after consulting with reviewers, we came to the conclusion that this would be beyond what could be expected with the turnaround time that we seek at *eLife* for acceptance of papers for publication.

*Reviewer #1:*

The authors have taken advantage of the properties of Arylsulfatase B (ARSB) to specifically degrade C4sulphate groups on GAG chains on inhibitory proteoglycans to promote regeneration in vitro but also in vivo using an optic nerve crush model. While the use of ARSB to promote regeneration is a novel and exciting idea given that this enzyme is already approved clinically, the paper contains a number of issues that are of concern.

1) In the Introduction, the authors mention that myelin is thought to be part of the "glial scar." Actually, myelin inhibitory proteins are located broadly and not thought to be specifically related to the scar.

2) Also, in the Introduction the authors champion their own work (which is reasonable) suggesting that C4S proteoglycans are those that are the most potently inhibitory. However, they do not even mention the strongly opposing and well published view from the Hsieh-Wilson lab that CS-E (rather than A) is critical. I think it would be best to mention up front that there is some controversy. Then, later, the authors can discuss how their current results help to provide evidence in support of their idea. However, it would have been welcomed if the authors had compared ARSB efficacy with CS-E specific degradation in their various models.

3) Figure 1 has a major problem in that the lengths of the control neurons with axons on untreated, TGF-b and TGF-b ARSB cultures are very similar, with very tiny differences (only around 10 um) between the groups. The same problem is apparent in the conditioned media experiments. However, the authors suggest that such small differences are statistically significant. I see no obvious differences between any of the neurons in Figure 1E suggesting that the differences really are incredibly slight. It would have been much preferable if an adult neuron (such as DRGs) would have been used and the cultures had been allowed to mature until long axons (in the controls) had been elaborated.

4) In Figure 2 the absolute levels of C4S in the optic nerve lesion is very small compared to the CS56 staining especially at day 7 which is supposedly at peak of expression. Indeed, the immunohistochemistry overall is not especially convincing. This suggests that C6S GAG moieties are in high abundance in the lesion and apparently in much greater abundance than C4S. Fold changes in C4S are not as interesting as are absolute amounts.

5) In Figure 3 there is a critical flaw. In the control optic crush animals there should be zero regeneration of axons into and past the lesion. None-the-less, the authors report regeneration of numerous axons in the controls as far as 1mm beyond the lesion. This is clear evidence of axonal sparing after lesion which confounds the interpretation of all of the data. In addition, the differences that are shown between control and ARSB treated optic nerves is again, incredibly small.

In its present form this manuscript should not be acceptable for publication.

*Reviewer #2:*

In their submission entitled "Identification of a critical sulfation in chondroitin that inhibits axonal regeneration" Pearson et al. examine the role of terminal 4S groups on GAG chains of CSPGs in imparting growth inhibiting properties of CSPGs, using the enzyme arylsulfatase B. They first show in astrocyte and neuronal coculture, or neuronal culture exposed to astrocyte conditioned medium, that ARSB performs similarly to chondroitonase ABC in restoring axonal growth potential after TGF-β induced inhibition. They then characterize an optic nerve crush model to show that 4S CSPGs are a part of the scar that forms after injury, which is present for at least 4 weeks post crush. Then the authors combine a zymosan+cAMP combinatorial treatment to induce RGC regeneration with ChABC or ARSB and show that RGC axon regeneration is improved similarly between the two treatments and can act on a delay of up to three days and does not alter astrocytic distributions.

While the finding that, in this model system, targeting 4S terminal GAG chains for cleavage can mediate effects similar to that of ChABC with an already clinically approved drug is quite interesting, overall the chosen application seems inappropriate. In the introductory sections, the authors make a strong point that ARSB is already approved for clinical use, but then they apply it to a system where ChABC itself does nothing without a combinatorial therapy that would not be seen in clinics. Without the application of this suggested therapy to an injury model where beneficial results would be expected sans untranslatable combinatorial therapies (eg spinal cord injury) a major thrust of this study is lost. I would highly recommend considering a different system for this application.

Additionally, although the authors were able to demonstrate that their delivery of ChABC was able to penetrate the optic nerve to some extent and breakdown CSPGs to some extent, they do not examine the efficiency very clearly. How much of the intact CSPGs remain should be demonstrated by immunostaining, and while it is appreciably difficult to clearly show ARSB function in vivo, it needs to be shown that the treatment worked as intended. Is there no biochemical approach to examining the GAG terminal chains?

*Reviewer #3:*

This is an important paper. It is well done, thoughtful about controls and relatively well versed in glycobiology. The results of these complicated animal studies, especially, are clearly presented and the overall results are unexpected. The rationales for carrying on further work using ARSB are clear and these initial results are hopeful.

I have no objection to the general direction, but I wonder about the need for the zymosan injection to prime the system in some way. What would be the equivalent in a patient? or what other insults would occur in a therapeutic setting? Clearly, they would hope this approach is something general and not only restricted to ONC. I think it is a well worthwhile study.

All that said, I do have some suggestions for improvement. First, I don't accept their claim of a 2.5 fold increase in 4S (antibody 2H6) as shown in Figure 2—figure supplement 1. The level at Day "0" is already ~1.75 fold (not baseline of 1.0) and it is shown as an increase to 2.5 fold. That's only a 1.4 fold increase, which is about the same as the CS-56 antibody. So, there is no selectivity. It seems misleading to state that. If I have misinterpreted the data, I'd like to see their justification for the claim based on these results.

In Figure 1, there are so few tubulin staining cells. I'd like to see more pictures, perhaps a small composite. How many of these cells were actually counted? How many were in the "longest" category. Also, it would be helpful to see the distribution of lengths, not only the (few?) longest ones. I can't tell based on the presentation. So, I have some skepticism.

[Editors’ note: what now follows is the decision letter after the authors submitted for further consideration.]

Thank you for submitting your article "Identification of a critical sulfation in chondroitin that inhibits axonal regeneration" for consideration by *eLife*. Your article has been reviewed by three peer reviewers, and the evaluation has been overseen by Joseph Gleeson as the Reviewing Editor and Marianne Bronner as the Senior Editor. Two of the three reviewers were the same and one reviewer is new for this submission. The following individual involved in review of your submission has agreed to reveal his identity: Hudson H Freeze (Reviewer #3).

The reviewers have discussed the reviews with one another and the Reviewing Editor has drafted this decision to help you prepare a revised submission.

Summary:

This manuscript leverages off of previous findings that implicated 4-sulfation as a mechanism of CSPG inhibition of axonal growth. The three reviewers agree that the major finding presented are that the human enzyme arylsulfatase B (ARSB), which cleaves 4-sulfate groups from glycosaminoglycan (GAG) chains of chondroitin sulfate proteoglycans (CSPGs), abrogates the inhibitory actions of chondroitin sulfate in vitro and promotes regeneration of retinal ganglion cell axons in the optic nerve. The reviewers from the previous submission appreciate the positive changes and the correction to the manuscripts of the previous errors. Reviewers commented that the work is consistent with the report of Yoo et al., 2013 using ARSB together with ChABC that suggested a comparable impact of these enzymes on locomotor function recovery after spinal cord injury. The reviewers appreciated the quality of the cell culture and animal model data and the potential impact of ARSB in axon regeneration. They mention that the study is important and well thought out, with immediate therapeutic implications and possibly applications. However, there were several concerns expressed by the reviewers that require attention before the manuscript can be further considered at *eLife*.

Essential revisions:

1) Considering the known variation associated with zymosan treatment, it is questionable how much meaningful impact ARSB has on axon regeneration. In addition, in the image in Figure 3B, "labeled axons" in the distal optic nerve appear to be bundled, possibly spared axons.

2) 1- CS-56 staining in conditioned media is not altered by ARSB treatment. However, 2H6 antibody has more affinity for 4S groups versus 6S. It would be interesting to document how ARSB treatment impacts the elevation of chondroitin sulfate proteoglycan after optic nerve crush as documented in Figure 2.

3) The authors don't provide significant data to further incriminate 4S versus 4,6S role in the inhibitory effect of CSPGs as well as any new data to specifically illustrate the fundamental role of 4S. A Title reflecting the effect of ARSB enzyme in their specific ONC paradigm will be more appropriate.

4) The representation of the data related to the neurite outgrowth does not allow assessing easily the inhibitory effect of CSPGs. Several experiments were performed in Figure 1 and Figure 1—figure supplement 1; however different representations of the data are used. Additionally, the authors should provide standard deviations to assess the reproducibility between triplicates.

5) The authors should provide information on the amount of enzymes used in subsection “Modifying CSPG sulfation enhances retinal ganglion cell axon regener”. One is expressed as U/ml and others as ug/ml. Please include both, even though the assays are based on rather artificial substrate measurements.

6) In subsection “Sulfation dictates the effects of CSPGs on axon growth” says that ARSB cleaves extracellular GAG chains, but only an assay using an artificial substrate is presented. Besides, the enzyme should not cleave GAG chains- it cleaves terminal 4-sulfate. That statement should be corrected or explained.

7) No data was shown on the effects of ARSB on 2H9 staining/reactivity. Presumably there is not any, since it only removes 1 sulfate per CS chain, but this should be stated explicitly.

8) The size of the CS chains in these cells/settings is not known. Presumably its 50-100 monosaccharides, but again ARSB should only remove a single 4-sulfate. This remarkably small change (1/20, 1/50, 1/100) is worth stressing, and citing relevant literature.

---

## [Author Response]

[Editors’ note: the author responses to the first round of peer review follow.]

This current submission has been extensively revised in response to the reviewers’ comments on our original submission. We have readily corrected two key errors (noted in the bullets below), and have additionally revised the manuscript to answer each of the other major and minor criticisms. We hope that this document summarizing our changes will persuade you to consider our revised manuscript for publication.

Having carefully read each of the reviewers’ comments, we noted that two of the major flaws highlighted by reviewers were, in fact, due to our figure presentation, rather than experimental design. Briefly, these included:

1) An omission in Figure 3 led Reviewer #1 to suggest that our optic nerve samples were detecting spared axons in negative control samples. In fact, these were regenerating axons from a Zymosan-treated control. However, we mistakenly omitted the “Zymosan” label from one panel of our graph.

2) An omission in Figure 2–Figure Supplement 1 led Reviewer #3 to question the validity of our data showing that CSPGs are elevated after optic nerve crush. We had failed to include the baseline “Day 0” condition where fold change = 1.0. We have fixed both of these errors.

Of the remaining criticisms, we address the most significant first:

1) “Spared” axons in controls

Reviewer #1:

In Figure 3 there is a critical flaw. In the control optic crush animals there should be zero regeneration of axons into and past the lesion. None-the-less, the authors report regeneration of numerous axons in the controls as far as 1mm beyond the lesion. This is clear evidence of axonal sparing after lesion which confounds the interpretation of all of the data. In addition, the differences that are shown between control and ARSB treated optic nerves is again, incredibly small.

The appearance of axonal sparing was due to our mislabeling of the graph in panel (c) of Figure 3, where we failed to specify that all animals in this experiment were treated with Zymosan/CPT-cAMP. This information was conveyed in the text and figure legend, but mistakenly omitted from the graph labels. We have corrected the figure. Our data confirming that no axons extended beyond the lesion in untreated control animals can be found in Figure 3—figure supplement 1. We therefore argue that this “critical flaw” was merely a labeling omission and does not in any way confound the interpretation of the data.

Regarding the differences shown between control and ARSB treated optic nerves, we believe our reported differences are substantial and noteworthy. The enhancement of axon regeneration in Zymosan + ARSB vs. Zymosan + buffer was on the order of ~250 axons at 0.50 mm from the lesion after only 14 d (Figure 3C). The inclusion of ARSB essentially doubles the effect of Zymosan alone and is comparable to the effects achieved in many landmark publications in the field of optic nerve regeneration. For instance, Lim et al., (2016) also used a combination stimulus and, in several experiments, reported effects of a similar magnitude to ours (e.g. in Lim et al., Figure 3, treatment group 3 = ~75 axons and group 4 = ~275 axons at 0.50 mm).

2) Use of intrinsic stimulus

Reviewer #2:

While the finding that, in this model system, targeting 4S terminal GAG chains for cleavage can mediate effects similar to that of ChABC with an already clinically approved drug is quite interesting, overall the chosen application seems inappropriate. In the introductory sections, the authors make a strong point that ARSB is already approved for clinical use, but then they apply it to a system where ChABC itself does nothing without a combinatorial therapy that would not be seen in clinics. Without the application of this suggested therapy to an injury model where beneficial results would be expected sans untranslatable combinatorial therapies (eg spinal cord injury) a major thrust of this study is lost. I would highly recommend considering a different system for this application.

Reviewer #3:

I have no objection to the general direction, but I wonder about the need for the zymosan injection to prime the system in some way. What would be the equivalent in a patient? or what other insults would occur in a therapeutic setting? Clearly, they would hope this approach is something general and not only restricted to ONC. I think it is a well worthwhile study.

We appreciate that both reviewers acknowledge the interest and significance of the work and welcome the opportunity to defend our choice of the optic nerve system and our use of a combined intrinsic/extrinsic regeneration stimulus. The optic nerve has many advantages for the study of axonal regeneration as compared to spinal cord injury models, which have enabled this work. The surgical accessibility of the optic nerve enables a high degree of reproducibility and allowed us to deliver ARSB directly to the lesion site. All axons in the optic nerve come from retinal ganglion cells, making the measurement of regeneration unambiguous.

Virtually all experimental therapies with robust effects in the optic nerve or spinal cord require combinatorial treatments to stimulate significant axon regeneration. While we appreciate that Zymosan itself could not be implemented clinically, ARSB in combination with other stimuli is a distinct possibility. Reporting the efficacy of ARSB in enhancing intrinsic pro-regenerative stimuli will certainly encourage other groups to integrate ARSB into new experimental therapies. We ourselves are engaged in ongoing research to this effect. We believe that holding back our findings with ARSB due to its failure to achieve an effect independently of intrinsic stimulation is not justified, given the state of the field and the potential impact of our findings.

3) CSPG immunohistochemistry

Reviewer #1:

In Figure 2 the absolute levels of C4S in the optic nerve lesion is very small compared to the CS56 staining especially at day 7 which is supposedly at peak of expression. Indeed, the immunohistochemistry overall is not especially convincing. This suggests that C6S GAG moieties are in high abundance in the lesion and apparently in much greater abundance than C4S. Fold changes in C4S are not as interesting as are absolute amounts.

We believe that the impression of low C4S levels may have been due to the false-coloring of our figures – the magenta coloring for 2H6 is more difficult to see on a black background than the green coloring of CS-56 when comparing them side by side. We have thus modified Figure 2 to show all images in grayscale, where features are more readily detected by eye. For the sake of full transparency, we have also modified the Figure 2—figure supplement 1 to include both raw fluorescence intensity values and fold change. These graphs are essentially identical, showing increase in both total GAG (using CS-56) and 4S GAG (using 2H6) as compared with the uninjured control in both analyses.

We appreciate reviewer #1’s suggestion that we compare absolute levels of CSPG and C4S, as this would indeed resolve the question of whether C4S or C6S moieties are in higher abundance within the lesion. However, it is not possible to compare absolute levels of two different antigens using immunohistochemistry, as each antibody has a unique epitope and affinity for the antigen, and the images are collected in different confocal channels with distinct gain and contrast settings. To quantify absolute amounts of GAG would require biochemical methods, which is precluded by the small amount of tissue in each optic nerve, as isolating the optic nerve lesion site produces only 6-8 µg of protein per sample and would require a prohibitive number of mice. We therefore chose to report fold change of total GAG and 4S GAG as compared to control.

Reviewer #2:

In their submission entitled "Identification of a critical sulfation in chondroitin that inhibits axonal regeneration" Pearson et al. examine the role of terminal 4S groups on GAG chains of CSPGs in imparting growth inhibiting properties of CSPGs, using the enzyme arylsulfatase B. They first show in astrocyte and neuronal coculture, or neuronal culture exposed to astrocyte conditioned medium, that ARSB performs similarly to chondroitonase ABC in restoring axonal growth potential after TGF-β induced inhibition. They then characterize an optic nerve crush model to show that 4S CSPGs are a part of the scar that forms after injury, which is present for at least 4 weeks post crush. Then the authors combine a zymosan+cAMP combinatorial treatment to induce RGC regeneration with ChABC or ARSB and show that RGC axon regeneration is improved similarly between the two treatments and can act on a delay of up to three days and does not alter astrocytic distributions.

We agree and have therefore modified the text to simply describe an increase in 4S and removed any claims regarding disproportionate expression. Our previous publications have documented a relative increase in 4S expression after injury as evaluated by biochemical methods. However, as stated above, these biochemical measurements are impossible to perform at the site of an optic nerve lesion given the extremely small amount of tissue.

Reviewer #3:First, I don't accept their claim of a 2.5 fold increase in 4S (antibody 2H6) as shown in Figure 2—figure supplement 1. The level at Day "0" is already ~1.75 fold (not baseline of 1.0) and it is shown as an increase to 2.5 fold. That's only a 1.4 fold increase, which is about the same as the CS-56 antibody. So, there is no selectivity. It seems misleading to state that. If I have misinterpreted the data, I'd like to see their justification for the claim based on these results.

This impression arose from our failure to include the baseline condition (1.0) at Day 0, instead starting our data from Day 1 (where the fold change was indeed ~1.75). We are grateful for the correction and have amended the figure to include “Day 0” baseline data for clarity.

4) Small differences in neurite length

Reviewer #1:

Figure 1 has a major problem in that the lengths of the control neurons with axons on untreated, TGF-b and TGF-b ARSB cultures are very similar, with very tiny differences (only around 10 um) between the groups. The same problem is apparent in the conditioned media experiments. However, the authors suggest that such small differences are statistically significant. I see no obvious differences between any of the neurons in Figure 1E suggesting that the differences really are incredibly slight. It would have been much preferable if an adult neuron (such as DRGs) would have been used and the cultures had been allowed to mature until long axons (in the controls) had been elaborated.

Reviewer #3:

In Figure 1, there are so few tubulin staining cells. I'd like to see more pictures, perhaps a small composite. How many of these cells were actually counted? How many were in the "longest" category. Also, it would be helpful to see the distribution of lengths, not only the (few?) longest ones. I can't tell based on the presentation. So, I have some skepticism.5) Efficiency of enzyme delivery

Reviewer #2:

*Additionally, although the authors were able to demonstrate that their delivery of ChABC was able to penetrate the optic nerve to some extent and breakdown CSPGs to some extent, they do not examine the efficiency very clearly. How much of the intact CSPGs remain should be demonstrated by immunostaining. And while it is appreciably difficult to clearly show ARSB function* in vivo*, it needs to be shown that the treatment worked as intended. Is there no biochemical approach to examining the GAG terminal chains?*

We appreciate the suggestion and agree that quantitatively demonstrating ARSB function in vivo would be ideal. However, absolute levels of GAG can only be obtained biochemically, which is limited by the amount of tissue in the injured optic nerve. To answer the question would take prohibitively high numbers of animals per condition, which is inconsistent with current pressures to reduce unnecessary use of animals in research.

Using immunohistochemistry, we found a significant increase in BE-123 “stub” immunoreactivity, indicative of ChABC activity (Figure 3—figure supplement 2), and a small but not significant decrease in CS-56 immunoreactivity at 14 days post crush. We also recovered the implanted gelfoam scaffolds from enzyme-treated mouse optic nerves and detected significant residual ARSB activity from ARSB-loaded scaffolds when compared with buffer-loaded scaffolds, which showed no ARSB activity. Taken together with the pro-regenerative effects seen in enzyme-treated mice, we believe these observations adequately support our conclusion that ChABC and ARSB were active and functioning in vivo.

[Editors' note: the author responses to the re-review follow.]

Essential revisions:1) Considering the known variation associated with zymosan treatment, it is questionable how much meaningful impact ARSB has on axon regeneration. In addition, in the image in Figure 3B, "labeled axons" in the distal optic nerve appear to be bundled, possibly spared axons.

We agree that analyzing optic nerve samples for the presence of spared axons is essential to ensure the validity of the results, including careful adherence to such guidelines as those recently published by Fischer et al.,(2017). We carefully reexamined the GAP-43-labeled axons in each image quantified from the sample shown in Figure 3B, and confirmed that no signs of bundled, excessively straight, obviously spared axons were present in any of the other sections analyzed. We also stained optic nerve samples for GFAP and found a clear injury site demarcated by retracted astrocyte processes. Further, we observed that, unlike CTB, which will appear in non-lesioned or spared axons if they are present, GAP-43 stains only regenerating axons and does not stain intact, non-lesioned axons. We have added a sentence to this effect in the manuscript and cited previously published work that reports the same findings. Collectively these observations give us confidence that our analysis and results include only regenerating axons.

2) 1- CS-56 staining in conditioned media is not altered by ARSB treatment. However, 2H6 antibody has more affinity for 4S groups versus 6S. It would be interesting to document how ARSB treatment impacts the elevation of chondroitin sulfate proteoglycan after optic nerve crush as documented in Figure 2.

We appreciate this suggestion and have added a new supplementary figure (Figure 3—Figure supplement 5) to address the question. We show that treating lesioned mouse optic nerves with ARSB at 3 dpc does not significantly alter immunoreactivity of CS-56 or 2H6 at 7 dpc when compared with the buffer-treated control.

3) The authors don't provide significant data to further incriminate 4S versus 4,6S role in the inhibitory effect of CSPGs as well as any new data to specifically illustrate the fundamental role of 4S. A Title reflecting the effect of ARSB enzyme in their specific ONC paradigm will be more appropriate.

The only known action of ARSB is to remove the sulfate group from the 4-position of GalNAc. This confirms the essential role of 4S (even if it is in the presence of 6S) as an important signal to neurons. Whether ARSB attacks 4,6S is not at all known, and we found a very low level of 4,6S at the non-reducing end of CSPGs by our HPLC methods. Based on this we feel that the title to the manuscript is appropriate.

4) The representation of the data related to the neurite outgrowth does not allow assessing easily the inhibitory effect of CSPGs. Several experiments were performed in Figure 1 and Figure 1—figure supplement 1; however different representations of the data are used. Additionally, the authors should provide standard deviations to assess the reproducibility between triplicates.

We have changed all data representations to dot plots, which we believe to be a more accurate picture of the data presented. Each experiment showed a significant action of both CSPGs and ARSB in reversing the CSPG actions on neurite outgrowth. We have included a Table of in vitro experimental statistics below containing the mean, standard deviation, standard error, and *p*-values (for both overall ANOVA and posthoc comparison to untreated samples) for each experiment in the triplicate of the in vitro experiments.

5) The authors should provide information on the amount of enzymes used in subsection “Modifying CSPG sulfation enhances retinal ganglion cell axon regener”. One is expressed as U/ml and others as ug/ml. Please include both, even though the assays are based on rather artificial substrate measurements.

We agree and have standardized our reporting of enzyme amounts, using mg/mL or µg/mL units for both enzymes.

6) In subsection “Sulfation dictates the effects of CSPGs on axon growth” says that ARSB cleaves extracellular GAG chains, but only an assay using an artificial substrate is presented. Besides, the enzyme should not cleave GAG chains--it cleaves terminal 4-sulfate. That statement should be corrected or explained.

We agree and have amended the sentence to more clearly state that ARSB cleaves terminal 4S groups from the non-reducing ends of GAG chains.

7) No data was shown on the effects of ARSB on 2H9 staining/reactivity. Presumably there is not any, since it only removes 1 sulfate per CS chain, but this should be stated explicitly.

We have included this data, showing no significant change in 2H6 immunoreactivity following ARSB treatment, in a new supplementary figure (Figure 3—figure supplement 5).

8) The size of the CS chains in these cells/settings is not known. Presumably its 50-100 monosaccharides, but again ARSB should only remove a single 4-sulfate. This remarkably small change (1/20, 1/50, 1/100) is worth stressing, and citing relevant literature.

We appreciate this suggestion and have added to our Discussion section emphasizing the fact that ARSB only removes a single, terminal sulfate from the non-reducing end of GAG chains, and yet produces a pro-regenerative effect of the same magnitude as that of ChABC, which digests entire GAG chains. We have cited literature on the average length of GAG chains for neurocan and phosphacan that shows an average chain length of approximately 50 disaccharides, which implies a small change in sulfation produces a big change in inhibitory activity.